# Cardiac Hypertrophy Changes Compartmentation of cAMP in Non-Raft Membrane Microdomains

**DOI:** 10.3390/cells10030535

**Published:** 2021-03-03

**Authors:** Nikoleta Pavlaki, Kirstie A. De Jong, Birgit Geertz, Viacheslav O. Nikolaev, Alexander Froese

**Affiliations:** 1Institute of Experimental Cardiovascular Research, University Medical Center Hamburg-Eppendorf, 20246 Hamburg, Germany; n.pavlaki@uke.de (N.P.); k.de-jong@uke.de (K.A.D.J.); a.froese@uke.de (A.F.); 2German Center for Cardiovascular Research (DZHK), Partner Site Hamburg/Kiel/Lübeck, 20246 Hamburg, Germany; b.geertz@uke.de; 3Institute of Experimental Pharmacology and Toxicology, University Medical Center Hamburg-Eppendorf, 20246 Hamburg, Germany

**Keywords:** cAMP, biosensor, cardiac hypertrophy, non-raft sarcolemma, compartmentation, fluorescence resonance energy transfer, microdomain, phosphodiesterase

## Abstract

3′,5′-Cyclic adenosine monophosphate (cAMP) is a ubiquitous second messenger which plays critical roles in cardiac function and disease. In adult mouse ventricular myocytes (AMVMs), several distinct functionally relevant microdomains with tightly compartmentalized cAMP signaling have been described. At least two types of microdomains reside in AMVM plasma membrane which are associated with caveolin-rich raft and non-raft sarcolemma, each with distinct cAMP dynamics and their differential regulation by receptors and cAMP degrading enzymes phosphodiesterases (PDEs). However, it is still unclear how cardiac disease such as hypertrophy leading to heart failure affects cAMP signals specifically in the non-raft membrane microdomains. To answer this question, we generated a novel transgenic mouse line expressing a highly sensitive Förster resonance energy transfer (FRET)-based biosensor E1-CAAX targeted to non-lipid raft membrane microdomains of AMVMs and subjected these mice to pressure overload induced cardiac hypertrophy. We could detect specific changes in PDE3-dependent compartmentation of β-adrenergic receptor induced cAMP in non-raft membrane microdomains which were clearly different from those occurring in caveolin-rich sarcolemma. This indicates differential regulation and distinct responses of these membrane microdomains to cardiac remodeling.

## 1. Introduction

3′,5′-Cyclic adenosine monophosphate (cAMP) is a ubiquitous second messenger which plays a crucial role in heart function and disease. Multiple G-protein coupled receptors including β-adrenergic receptors (β-ARs) and prostaglandin EP receptors can stimulate cAMP production in cardiac myocytes, often leading to different functional responses [1,2,3,4]. This phenomenon can be explained by tight compartmentation of receptor-dependent cAMP signaling in spatially-confined subcellular micro- or nanodomains where receptors, cAMP, and its downstream effector proteins trigger key functional effects [4,5]. Recently, the development of genetically encoded biosensors based on Förster resonance energy transfer (FRET) have enabled cAMP measurements in intact living cells including adult ventricular cardiomyocytes [6,7,8,9,10].

Generation of compartmentalized cAMP responses starts at the membrane of cardiomyocytes, which contains at least two types of microdomains. One can be formed by caveolae, a specific subset of lipid rafts associated with the scaffolding protein caveolin, and another one has been suggested to reside in non-raft microdomains, with clear biochemical and functional evidence of their different roles in the regulation of receptor mediated cAMP generation and downstream signaling [11,12,13]. Using newly developed FRET-based biosensors targeted to lipid raft and non-raft domains of the plasma membrane, differences in basal and β-AR or prostaglandin stimulated cAMP levels associated with these microdomains were initially detected in HEK293 cells [14], and later also in healthy rat myocytes [15]. However, precious little is still known about the disease-driven changes occurring in the non-raft sarcolemmal microdomains. This is in contrast to the caveolin-rich membrane microdomain, for which we have previously developed transgenic mice with AMVM expression of the specifically targeted biosensor pmEpac1. When subjected to a pressure overload model of cardiac hypertrophy induced by transverse aortic constriction (TAC), this mouse model uncovered a disease-associated redistribution of cAMP degrading phosphodiesterases (PDEs), PDE2 and PDE3 between microdomains regulating β_1_- and β_2_-AR induced cAMP generation and receptor-dependent contractile responses [16].

To study disease-driven alterations of cAMP signals in non-caveolar sarcolemma, here we generated a novel transgenic mouse line expressing a highly sensitive FRET-based biosensor E1-CAAX targeted to non-raft membrane microdomains and subjected them to the same TAC model. We could detect specific changes in β-AR subtype responses and PDE3-dependent compartmentation of β_1_- and β_2_-AR induced cAMP which indicate differential regulation of local cAMP in caveolin-rich and non-raft membrane microdomains in response to cardiac remodeling.

## 2. Materials and Methods

### 2.1. Cloning and Transgenic Mouse Generation

DNA encoding the CAAX box sequence (KKKKKSKTKCVIM) from Rho GTPase was attached to the carboxyl terminus of Epac1-camps sensor [17] via BamHI restriction site (Figure 1a), this sequence enables specific targeting of proteins to non-lipid raft domains of the plasma membrane [14,18]. The resulting E1-CAAX sensor sequence was further subcloned into the previously described vector containing the α-myosin heavy chain (αMHC) promoter and simian virus (SV40) polyadenylation signal [9], which was then linearized with SpeI, purified, and used for pronuclear injections to generate transgenic mice as previously described [16]. Founder mice and their heterozygote offspring were kept on FVB/N background and genotyped by a standard PCR using the primers 5′-TGACAGACAGATCCCTCCTAT-3′ and 5′-CATGGCGGACTTGAAGAAGT-3′, resulting in a ~340 b.p. fragment on a gel.

### 2.2. Transverse Aortic Constriction (TAC)

9–12-week-old female mice were randomized into sham or TAC group. Mice were anesthetized using 1.5–2% isoflurane in 100% oxygen. A suprasternal incision was made, and the aortic arch was visualized using a binocular microscope (Olympus, Hamburg, Germany). TAC occurred by spacer defined (27-gauge) constriction using a 6–0 polyviolene suture between the first and second trunk of the aortic arch [19]. For the sham group, the aorta was exposed but not constricted. Three days after surgery, Doppler velocity was measured by a 20 MHz probe to quantify the pressure gradient across the TAC region or after sham procedure by transthoracic echocardiography (VisualSonics Vevo 3100; Toronto, ON, Canada). Mice received intraperitoneal analgesic therapy with buprenorphine and carprofen for three days post-surgery. Echocardiography was performed eight weeks after surgery with subsequent heart and AMVM isolation.

### 2.3. Adult Mouse Ventricular Myocyte Isolation and Live Cell Imaging

AMVMs were isolated from transgenic mice eight weeks after TAC or sham surgery by retrograde perfusion of the aorta with an enzymatic digestion solution in a Langendorff apparatus and plated on laminin-coated round glass cover slides for FRET measurements on the same day as described [20]. For FRET measurements, cover slides with adherent cells were mounted in an Attofluor microscopy chamber and maintained in FRET buffer containing 144 mmol/L NaCl, 5.4 mmol/L KCl, 1 mmol/L MgCl_2_, 1 mmol/L CaCl_2_, 10 mmol/L HEPES, pH 7.3. Live cell imaging was performed using a custom-made FRET microscopy system built around a Nikon Eclipse Ti microscope (Nikon, Düsseldorf, Germany) equipped with 63×/1.40 oil-immersion objective [21]. The donor fluorophore (CFP) was excited with a 440 nm light at 5 s interval using a CoolLED single-wavelength light emitting diode. Emitted light was separated into CFP and YFP channels using a DV2 DualView (Photometrics, Surrey, BC, Canada) and detected using an ORCA-03G charge-coupled device camera (Hamamatsu, Herrsching am Ammersee, Germany). After reaching a stable baseline, cells were treated with different compounds diluted in the FRET buffer to stimulate cellular cAMP responses. AMVMs from each animal have been tested on the same day with all drugs to better compare their effects. Offline data analysis was performed using ImageJ, Microsoft Excel, and GraphPad Prism 6 software as described [20]. To determine the E1-CAAX biosensor affinity, cells were first treated with 100 µmol/L of the adenylyl cyclase inhibitor MDL-12330A followed by increasing concentrations of the cell-permeable cAMP analogue 8-pCPT-2′-O-Me-cAMP-AM (BIOLOG Life Science Institute, Bremen, Germany) and analyzed using a previously established protocol [19].

### 2.4. Confocal Microscopy

Imaging was performed using Zeiss LSM 800 microscope (Carl Zeiss MicroImaging, Jena, Germany) equipped with a Plan-Apochromat 63×/1.40 oil-immersion objective. For co-localization experiments, cells were fixed for 5 min with ROTI Histofix^®^ 4% (Roth), washed, and stained overnight with mouse monoclonal caveolin 3 (BD #610421) antibody, followed by the secondary anti-mouse Alexa 633 Fluor^®^ antibody (A-21063, Life Technologies). Images were acquired for E1-CAAX sensor (488 nm diode laser) and for caveolin 3 (633 nm diode laser excitation) and automatically analyzed using the ZEN 2019 software (Zeiss, Jena, Germany).

### 2.5. Histology, Morphometric Analysis, and Echocardiography

Experiments were performed as previously described [22]. For cardiomyocyte dimension analysis, transverse heart sections were incubated with Wheat Germ Agglutinin (WGA, 75 μg/mL) for 30 min in the dark, washed thrice for 5 min with phosphate-buffered saline, mounted, and observed under a Keyence Fluorescence microscope Biozero BZ 8100. Images were acquired using Keyence Biozero imaging software (Keyence, Neu-Isenburg, Germany) and analyzed with ImageJ software. The cell area was measured in 50 cells from three individual hearts per group.

### 2.6. Immunoblot Analysis

Heart tissues were shock frozen and homogenized in a buffer containing: 10 mmol/L HEPES, 300 mmol/L sucrose, 150 mmol/L NaCl, 1 mmol/L EGTA, 2 mmol/L CaCl_2_ and 1% Triton-X. Proteins were quantified using Pierce BCA protein assay (Thermo Fischer Scientific, Dreieich, Germany). Samples were boiled at 70 °C for 10 min, and 15 μg of total protein per lane were subjected to 10% SDS-PAGE and to immunoblot analysis using anti-PDE2A antibody (Fabgennix, Frisco, TX, USA), custom-made rabbit polyclonal PDE3A antibody (kindly provided by Chen Yan, University of Rochester, Rochester, NY, USA), rabbit monoclonal PDE4B and PDE4D antibodies (Abcam, Berlin, Germany), EP4 receptor antibody (Santa Cruz Biotechnology, Heidelberg, Germany), rabbit polyclonal calsequestrin (Thermo Fischer Scientific, Dreieich, Germany), and mouse monoclonal anti-α-tubulin antibody (Sigma, Taufkirchen, Germany). All blots were scanned and analyzed densitometrically by ImageJ software for uncalibrated optical density.

### 2.7. Statistics

Normal distribution was tested by the Kolmogorov–Smirnov test, and differences between the groups of echocardiographic or morphometric data obtained from individual animals were analyzed using one-way ANOVA for simple two-group comparison or the Mann–Whitney test, as appropriate. FRET imaging data obtained using multiple cells isolated from several animals were analyzed using mixed ANOVA followed by Wald’s chi-squared test.

## 3. Results

### 3.1. E1-CAAX Biosensor Mouse Generation and Characterization

To monitor cAMP dynamics specifically in non-raft membrane microdomains of freshly isolated AMVMs, we generated a new transgenic mouse line expressing the E1-CAAX biosensor under the control of the AMVM specific αMHC promoter (Figure 1a). All isolated adult transgenic AMVMs expressed the sensor which was localized at the cell membrane (Figure 1b) as recently demonstrated in rat myocytes transduced with Epac2-CAAX biosensor adenovirus [15]. Additionally, as previously suggested, the affinity of the cAMP sensor was not significantly affected by the fusion of the CAAX box onto its C-terminus (Appendix A)

Transgenic mice at the age of three and six months had normal heart morphology and AMVM size comparable with wild-type littermates (Figure 1c–f). Additionally, echocardiography did not show any differences between transgenic mice and their wild-type control littermates in terms of morphometry and function (Table 1).

Importantly, transgenic mice also did not differ from their wild-type littermates in terms of the expression of major cardiac cAMP degrading PDEs—PDE2, PDE3, and PDE4, which could be confirmed by immunoblot analysis (Appendix A).

### 3.2. TAC Model

To study disease driven alterations in non-lipid raft membrane microdomains, we subjected E1-CAAX mice to a pressure overload induced cardiac hypertrophy model. Eight weeks after TAC, mice have developed a robust compensated cardiac hypertrophy phenotype without major reduction in contractility (Table 2), which is well documented for mice on the in FVB/N background [16,23,24,25,26]. In line with these previous studies, our compensated TAC model has not largely affected the whole-cell protein levels of cardiac PDE2, PDE3, and PDE4, as shown by immunoblots of heart lysates (Appendix A).

Importantly, E1-CAAX biosensor membrane localization was not affected by TAC (Figure 2a), despite hypertrophy, signs of fibrosis, and clear increase in cell size (Figure 2b–d).

Therefore, we could use this transgenic mouse model to study disease-driven alterations in local cAMP signals in non-caveolar membrane microdomains.

### 3.3. Live Cell Imaging of cAMP in Non-Lipid Raft Membrane Domains

Next, we isolated AMVMs from sham and TAC hearts eight weeks after surgery to perform live cells imaging of local microdomain-specific cAMP responses in freshly isolated cells.

#### 3.3.1. Responses to β-Adrenergic and Prostaglandin Receptor Stimulation

β-adrenergic receptor stimulation using the unselective β-agonist isoproterenol (ISO) led to a rapid decrease of FRET monitored as an increase in CFP/YFP ratio which is indicative of raising cAMP concentration in non-raft membrane microdomains. This response was strongly augmented by additional subsequent treatment with the non-selective PDE inhibitor 3-isobutyl-1-methylxanthine (IBMX), suggesting an important regulatory role of cardiac PDEs in these membrane microdomains (Figure 3a). Therefore, we went on to study the contribution of individual PDE families to the cAMP regulation in sham and TAC myocytes. Since cAMP signals induced by individual PDE inhibitors applied alone were negligible in E1-CAAX expressing cells (Appendix A), we further analyzed their effects after activation of individual cAMP stimulating receptor subtypes. 

Selective stimulation of β_1_-adrenergic receptors generated the strongest cAMP responses which were not affected by TAC. In contrast, β_2_-adrenergic receptors and prostaglandin E2 receptors triggered much smaller cAMP signals which were significantly increased in hypertrophied myocytes (Figure 3b), suggesting altered receptor expression or local PDE dependent regulation.

#### 3.3.2. PDE Dependent Regulation of β_1_-Adrenergic Receptor cAMP Responses

To study how local β_1_-AR/cAMP levels in non-raft microdomains are regulated by PDEs in healthy and diseased AMVMs, we stimulated cells freshly isolated from sham and TAC mice with either maximal (100 nmol/L) of submaximal (3 nmol/L) concentration of ISO [15,27] in presence of the β_2_-AR antagonist followed by selective inhibitors of PDE2, PDE3 and PDE4 families. Upon maximal receptor stimulation, all three PDE families were involved in cAMP hydrolysis with PDE4 being the predominant one, followed by PDE3 and PDE2. However, both sham and TAC groups showed comparable responses to the inhibition of PDE2 with BAY60-7550, PDE3 with cilostamide and PDE4 with rolipram (Figure 4).

Since maximal β_1_-adrenergic receptor stimulation can generate relatively high amounts of cAMP which might override possible differences in PDE-dependent regulation, we next treated the myocytes with a submaximal concentration of ISO to compare PDE inhibitor responses. Interestingly, the submaximal receptor activation resulted in a smaller contribution of PDE4 and could unmask a significantly reduced PDE3 inhibitor effect in TAC vs. sham myocytes (Figure 5).

#### 3.3.3. Responses to PDE Inhibitors after β_2_-Adrenergic Receptor Stimulation

To further study how cAMP levels are regulated by PDEs after β_2_-AR stimulation, we stimulated sham and TAC myocytes with 100 nmol/L of ISO in the presence of the β_1_-AR blocker CGP-20712A followed by PDE inhibitors as described above. Interestingly, similar to responses measured in caveolin-rich membrane microdomains [16], β_2_-AR/cAMP was strongly regulated by PDE3 and this effect was significantly diminished after TAC, whereas the response to other PDE inhibitors were not significantly changed in diseased cells (Figure 6).

#### 3.3.4. PDE Regulation of Prostaglandin Receptor Responses

Given the specific nature of the biosensor targeted to detect cAMP primarily in non-lipid raft membrane microdomains, we also sought to assess cAMP regulation after prostaglandin receptor stimulation which is known to specifically regulate these microdomains [14,15]. Although TAC led to somewhat higher PGE1 responses (see Figure 3b), we could not detect any significant changes in the contribution of individual PDE families in the regulation of these signals in diseased myocytes (Figure 7). Possible explanation for this results could be an increased expression of the relevant prostaglandin EP2/4 receptors after TAC. Indeed, immunoblots analysis with a specific EP4 receptor antibody revealed its clear upregulation in TAC heart lysates (Appendix A).

## 4. Discussion

Myocytes have provided a paradigm of a cell type with highly compartmentalized cAMP signaling [4,5,28]. Even within the plasma membrane itself there are at least two differentially regulated types of cAMP microdomains associated with caveolin-rich and non-raft sarcolemma. While specific disease-driven changes in cAMP compartmentation in caveolin-rich membrane have previously been investigated [16], no studies have addressed this issue for non-raft membrane microdomains.

To study real-time cAMP dynamics in non-raft microdomains of healthy and diseased AMVMs, we generated a new transgenic mouse model expressing the E1-CAAX biosensor in adult myocardium. This allows live cell imaging in freshly isolated AMVMs without the need to introduce FRET biosensors using, e.g., adenoviral gene transfer during prolonged ex vivo culture. Importantly, transgenic expression of the biosensor in mice did not lead to any morphological or functional abnormalities in their hearts (Figure 1, Table 1).

Another clear advantage of transgenic biosensor mice is that this model enables studies of pathological alterations in cAMP microdomains by subjecting these mice to experimental in vivo models of cardiac disease. In this work, we used a well-established model of pressure overload induced cardiac hypertrophy following TAC. As previously shown for wild-type and biosensor expressing mice on the on FVB/N genetic background [16,23,24,25,26], TAC in E1-CAAX mice led to a compensated state of cardiac hypertrophy without pronounced reduction in contractile function (Table 2). This compensated state of cardiac disease is especially interesting in terms of new therapeutic strategies acting already at early stages because it is not accompanied by gross changes of β_1_-AR, adenylyl cyclase, and PDE expression and activity at the whole cell level which is typical for chronic heart failure [16] (Appendix A). Instead, using transgenic mice expressing the pmEpac1 biosensor specifically targeted to caveolin-rich membrane microdomains, we could show that TAC leads to redistribution of PDE2 and PDE3 between different microdomains which led to augmented β_1_-AR dependent and reduced β_2_-AR dependent cAMP signals and contractility [16]. However, the question remained whether such PDE redistribution takes place also in non-raft membrane domains.

In contrast to caveolin-rich membrane where TAC led to augmentation of local β_1_-AR and reduction of β_2_-AR responses due to PDE redistribution [16], E1-CAAX responded to TAC with augmented β_2_-AR/cAMP signals without significant alterations of β_1_-AR/cAMP signals (see Figure 3b). This was accompanied by a reduction in the measured PDE3 inhibitor effects, which was the most prominent PDE associated with β_2_-AR/cAMP regulation in both membrane microdomains and was recently implicated into the tight regulation of β_2_-AR/cAMP in general [29]. This is in line with the redistribution of PDE3A away from the sarcolemma after TAC which, at least in part due to isoform switch, has been shown to relocate to sarcoplasmic reticulum fractions [16]. However, in contrast to pmEpac1 sensor measurements we did not observe any change in PDE2 dependent regulation of the non-raft microdomain during cardiac hypertrophy (Figure 4, Figure 5, Figure 6 and Figure 7). This suggests that this PDE is more tightly associated with the adrenergic receptors located in the caveolin-rich sarcolemma and with the regulation of their cAMP signals (Figure 8). Lack of changes in PDE2 mediated response after TAC might be also a reason for an increase in β_2_-AR responses in non-caveolar membrane microdomains of diseased myocytes following local depletion of PDE3, which contrasts with a decrease of β_2_-AR/cAMP in caveolin-rich membrane microdomains where decrease of local PDE3 is accompanied by an increase of PDE2 contribution. Interestingly, we did not observe strong effects of PDE inhibitors applied alone, which is in contrast to the data obtained in rat cardiomyocytes using an adenoviraly-expressed, slightly more sensitive Epac2-CAAX biosensor [15]. However, we could detect a strong increase of local cAMP upon IBMX treatment (Appendix A), suggesting that multiple PDEs act in concert to control basal cAMP levels in non-caveolar membrane microdomains of AMVMs. Additionally, we did not observe major changes in PDE regulation of prostaglandin receptor signaling after TAC, apart from a slight increase of the EPR/cAMP signal amplitude measured without PDE inhibition which is most likely due to higher expression of these receptors after TAC (Figure 3b and Figure 7; Appendix A). Since ERP system is known to be segregated from the pools of cAMP involved in the regulation of cardiac contractility [2,30], this effect of disease on local cAMP levels stimulated by prostaglandin E1 might convey the effects of inflammatory processes and mediators on myocytes function. Future studies are needed to understand the underlying molecular mechanisms including the substrates and macromolecular complexes of the cAMP dependent protein kinase (PKA) type I which has been found to be activated specifically by EPR signaling as opposed to of β-AR axis which is linked to phosphorylation the regulation of calcium handling and contractile proteins via PKA type II [31]. One possible scenario is that caveolin-rich membrane microdomains are predominantly involved in the regulation of disease-driven changes of contractile function [16], whereas non-raft membrane microdomains are regulating functional responses to inflammatory mediators and potential new functions linked to PKA type I and its molecular scaffolds.

## 5. Conclusions

In conclusion, using a newly generated E1-CAAX biosensor expressing mouse model subjected to pressure overload cardiac hypertrophy we could detect specific changes of cAMP compartmentation in non-raft membrane microdomains which were clearly distinct from those occurring in caveolin-rich sarcolemma. This confirms the existence of at least two distinct membrane microdomains for cAMP and their differential responses to cardiac remodeling.

## Figures and Tables

**Figure 1 cells-10-00535-f001:**
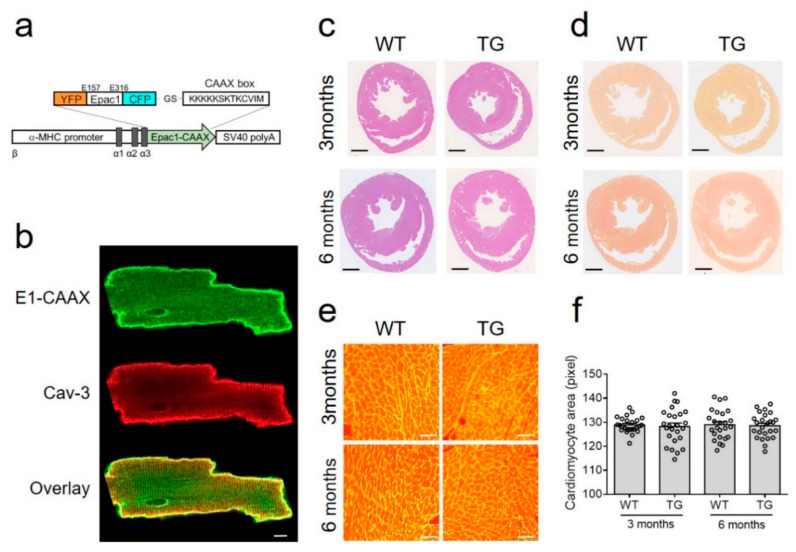
Generation and characterization of E1-CAAX transgenic mice. (**a**) The DNA encoding the CAAX box sequence (KKKKKSKTKCVIM) was attached to the C-terminus of Epac1-camps sensor and expressed under the control of the αMHC promoter followed by simian virus (SV40) polyadenylation signal. (**b**) Confocal images of isolated E1-CAAX adult transgenic AMVMs immunostained with caveolin 3 (Cav-3) antibody. Scale bar, 10 µm. (**c**) Hematoxylin and eosin and (**d**) picrosirius red stainings of cross-sections from three month old and six month old wild-type (WT) and transgenic (TG) mouse hearts. Scale bars, 1 mm. (**e**) Wheat germ agglutinin (WGA) staining of adult mouse ventricular cardiomyocytes in heart cross-sections. Scale bars, 50 µm. (**f**) Quantification of cardiomyocyte area from WT and TG mice at three months and six months of age.

**Figure 2 cells-10-00535-f002:**
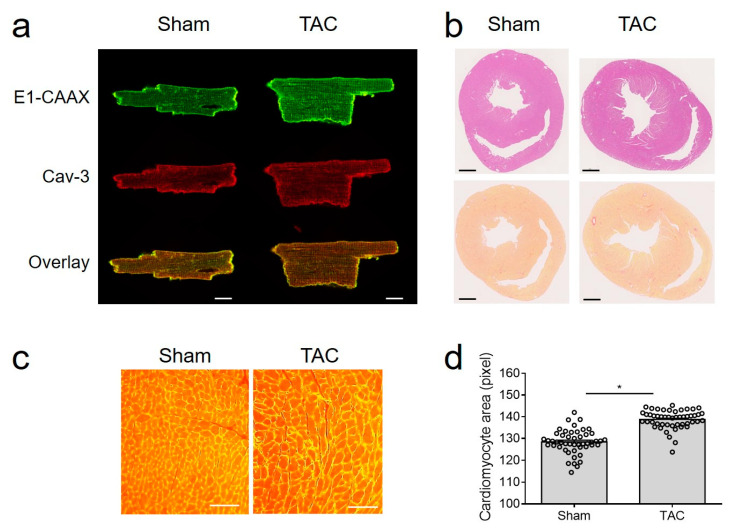
TAC model in E1-CAAX mice. (**a**) Confocal images of sham and TAC AMVMs taken as described in Figure 1b. Scale bar, 10 µm. (**b**) Hematoxylin/eosin and picrosirius red stainings of cross-sections from sham and TAC hearts of E1-CAAX mice. Scale bars, 1 mm. (**c**) WGA staining of sham and TAC heart tissues. Scale bars, 50 µm. (**d**) Quantification of AMVM area. * *p* < 0.05 by one-way ANOVA.

**Figure 3 cells-10-00535-f003:**
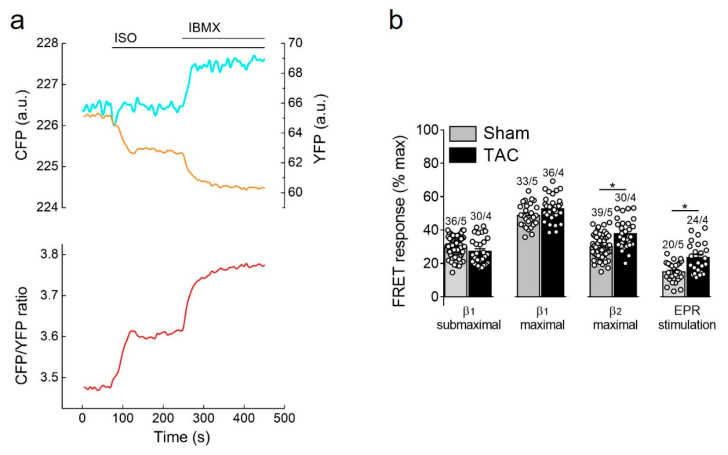
cAMP responses to β-adrenergic (β-AR) and prostaglandin receptor stimulation. (**a**) Representative single CFP and YFP intensities as well as CFP/YFP ratio trace (*n* = 10) recorded form a single E1-CAAX AMVM upon stimulation with isoproterenol (ISO, 100 nmol/L) and 3-isobutyl-1-methylxanthine (IBMX, 100 µmol/L). An increase in the FRET ratio represents an increase in local cAMP. (**b**) Amplitudes of cAMP response upon stimulation of β_1_-ARs (with submaximal 3 nmol/L or maximal 100 nmol/L concentration of ISO in the presence of the selective β_2_-AR blocker ICI118551, 50 nmol/L), β_2_-ARs (with 100 nmol/L ISO in the presence of the selective β_1_-AR blocker CGP20712A, 100 nmol/L) or prostaglandin EP2/4 receptors (EPR) with PGE1, 100 nmol/L. Responses to individual receptor stimulation were calculated from FRET ratio traces as a% of maximal response induced by forskolin (10 µmol/L) plus IBMX (100 µmol/L). Means ± SE. Number of cells/mice are stated above the bars. * *p* < 0.05 by mixed ANOVA followed by Wald’s chi-squared test.

**Figure 4 cells-10-00535-f004:**
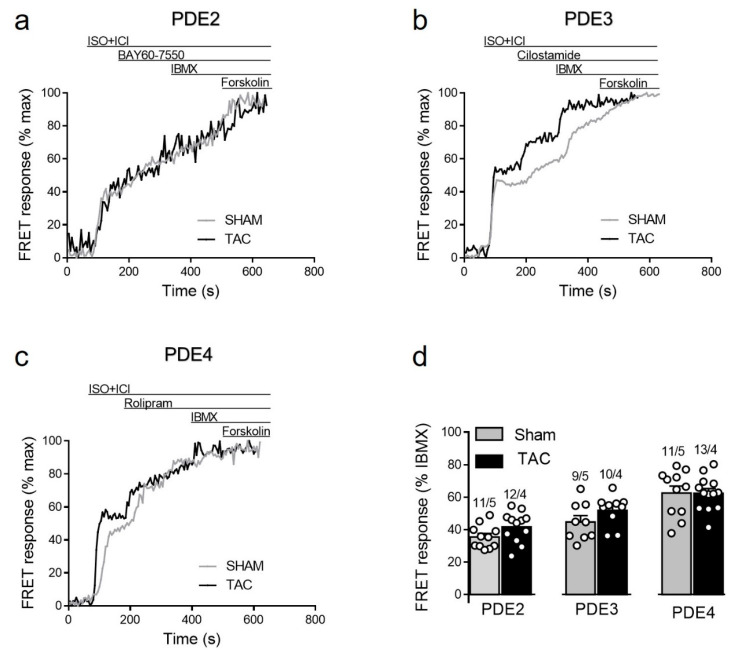
cAMP responses to selective PDE2, PDE3 and PDE4 inhibition after maximal β_1_-AR stimulation in sham and TAC myocytes. Representative FRET ratio traces recorded from cells in response to β_1_-AR stimulation with 100 nmol/L ISO in the presence 50 nmol/L ICI118551 (ISO+ICI), followed by (**a**) 100 nmol/L of the PDE2 inhibitor BAY60-7550, (**b**) 10 µmol/L of the PDE3 inhibitor cilostamide, and (**c**) 10 µmol/L of the PDE4 inhibitor rolipram. PDE inhibitor responses were calculated as the % maximal PDE inhibition with the subsequently applied non-selective PDE inhibitor IBMX (100 µmol/L). Forskolin (10 µmol/L) was applied at the end of each experiments to obtain the maximal possible FRET response. (**d**) Quantification of PDE inhibitor responses in sham and TAC myocytes. Means ± SE. Number of cells/mice are stated above the bars.

**Figure 5 cells-10-00535-f005:**
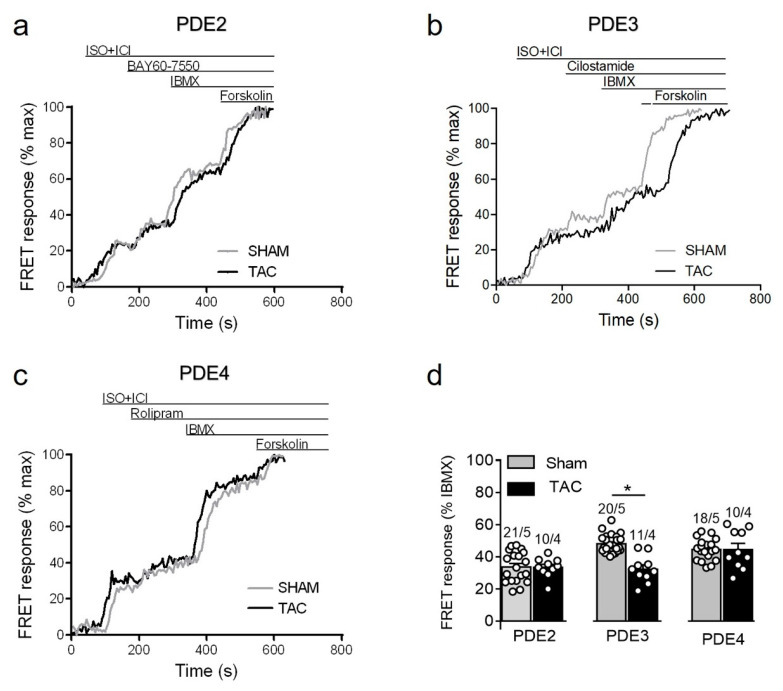
cAMP responses measured upon selective PDE2, PDE3, and PDE4 inhibition after submaximal β_1_-AR stimulation in sham and TAC myocytes. Representative FRET ratio traces recorded from cells in response to β_1_-AR stimulation with 3 nmol/L ISO in the presence 50 nmol/L ICI118551 (ISO+ICI), followed by (**a**) 100 nmol/L of the PDE2 inhibitor BAY60-7550, (**b**) 10 µmol/L of the PDE3 inhibitor cilostamide, and (**c**) 10 µmol/L of the PDE4 inhibitor rolipram. PDE inhibitor responses were calculated as a% maximal PDE inhibition by IBMX (100 µmol/L). Forskolin (10 µmol/L) was applied at the end of each experiments to obtain the maximal FRET response. (**d**) Quantification of PDE inhibitor responses in sham and TAC myocytes. Means ± SE. Number of cells/mice are stated above the bars. * *p* < 0.05 by mixed ANOVA followed by Wald’s chi-squared test.

**Figure 6 cells-10-00535-f006:**
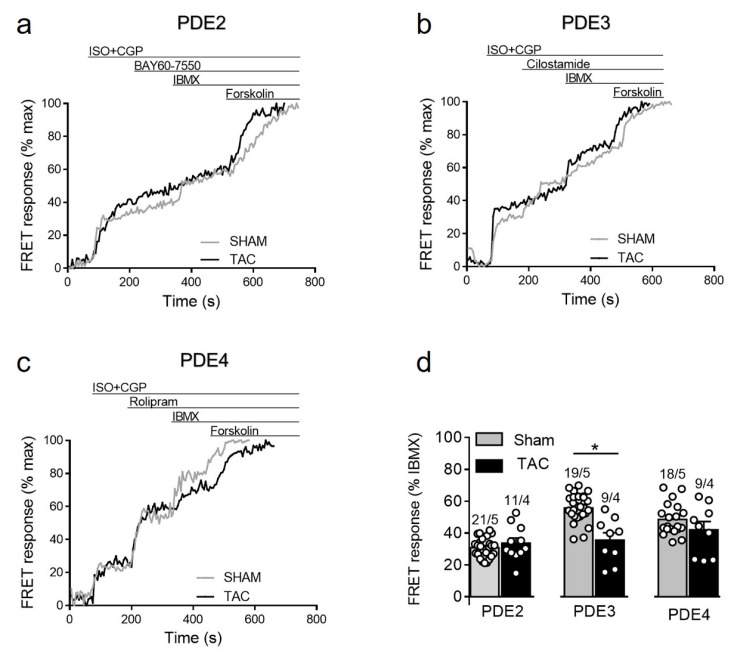
cAMP responses to selective PDE2, PDE3 and PDE4 inhibition after β_2_-AR stimulation in sham and TAC myocytes. Representative FRET ratio traces recorded from cells in response to β_2_-AR stimulation with 100 nmol/L ISO in the presence 100 nmol/L CGP20712A (ISO+CGP), followed by (**a**) 100 nmol/L of the PDE2 inhibitor BAY60-7550, (**b**) 10 µmol/L of the PDE3 inhibitor cilostamide, and (**c**) 10 µmol/L of the PDE4 inhibitor rolipram. PDE inhibitor responses were calculated as a% maximal PDE inhibition by IBMX (100 µmol/L). Forskolin (10 µmol/L) was applied at the end of each experiments to obtain the maximal FRET response. (**d**) Quantification of PDE inhibitor responses in sham and TAC myocytes. Means ± SE. Number of cells/mice are stated above the bars. * *p* < 0.05 by mixed ANOVA followed by Wald’s chi-squared test.

**Figure 7 cells-10-00535-f007:**
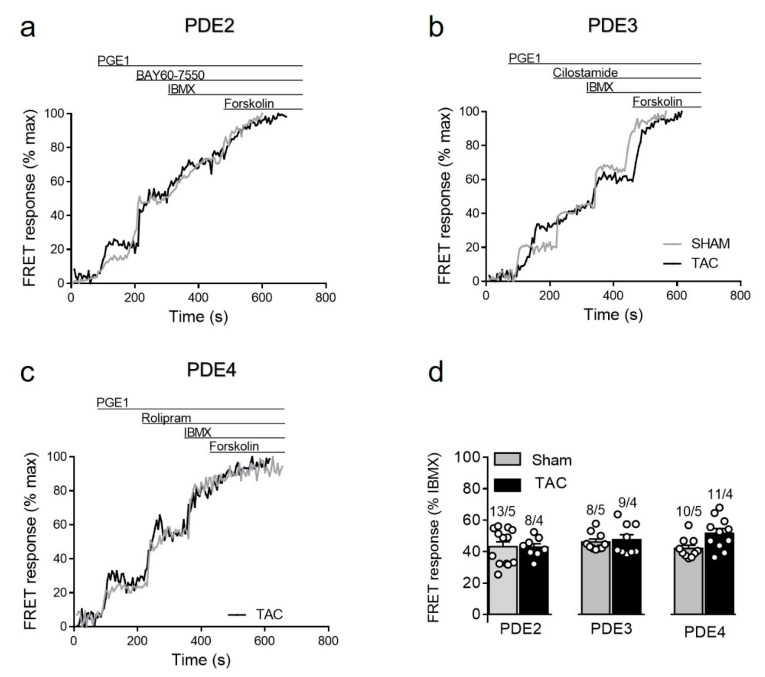
cAMP responses upon PDE inhibition after prostaglandin EP receptor stimulation in sham and TAC myocytes. Representative FRET ratio traces recorded from cells in response to EPR stimulation with 100 nmol/L of PGE1, followed by (**a**) 100 nmol/L of the PDE2 inhibitor BAY60-7550, (**b**) 10 µmol/L of the PDE3 inhibitor cilostamide, and (**c**) 10 µmol/L of the PDE4 inhibitor rolipram. PDE inhibitor responses were calculated as a% maximal PDE inhibition by 100 µmol/L IBMX). Forskolin (10 µmol/L) was applied at the end of each experiments to obtain the maximal FRET response. (**d**) Quantification of PDE inhibitor responses in sham and TAC myocytes. Means ± SE. Number of cells/mice are stated above the bars.

**Figure 8 cells-10-00535-f008:**
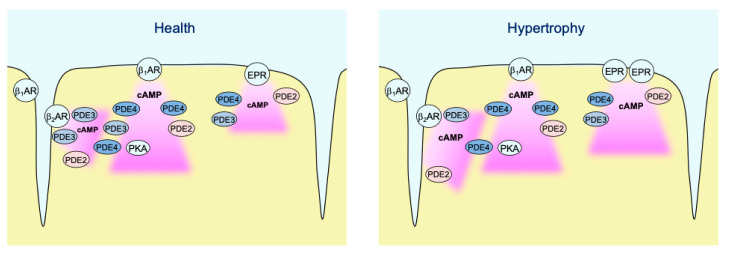
Schematic diagram highlighting the major findings of the study. In healthy AMVMs, cAMP synthesis in non-raft membrane microdomains can be stimulated by β_1_- and β_2_-adrenergic receptors (β-ARs), as well as by EP2/4 prostaglandin receptors (EPRs). Each pool of cAMP is under control by phosphodiesterases PDE2, PDE3, and PDE4. While all three PDEs are almost equally regulating EP2/4 receptor signals, β_1_-AR/cAMP is under predominant control by PDE4 and β_2_-AR/cAMP is mostly controlled by PDE3. In hypertrophy, PDE3 is redistributed away from β_2_-AR which leads to stronger local cAMP responses. In addition, EP2/4 receptor expression is upregulated after TAC, which leads to stronger cAMP responses to PGE1 even though the relative PDE effects of this pool of cAMP are not changed.

**Table 1 cells-10-00535-t001:** Echocardiographic parameters of Epac1-CAAX mice compared to wild-type littermates.

	Three Months Old		Six Months Old	
Parameter	Wild-Type	E1-CAAX	Wild-Type	E1-CAAX
IVDd (mm)	0.98 ± 0.04	1.00 ± 0.03	1.02 ± 0.04	1.03 ± 0.04
LVIDd (mm)	3.65 ± 0.06	3.60 ± 0.08	3.54 ± 0.11	3.50 ± 0.08
LVPWd (mm)	0.88 ± 0.04	0.96 ± 0.05	1.04 ± 0.08	1.10 ± 0.06
LV mass/BW (mg/g)	4.28 ± 0.18	4.39 ± 0.121	4.23 ± 0.18	4.19 ± 0.17
FS (%)	34.7 ± 0.8	36.7 ± 1.2	32.9 ± 1.9	36.0 ± 1.5
EF (%)	64.7 ± 1.1	67.3 ± 1.7	61.9 ± 2.9	66.5 ± 2.0
Stroke volume (μL)	37.3 ± 1.3	35.8 ± 1.7	33.7 ± 2.9	34.6 ± 2.3
Cardiac Output (mL/min)	16.9 ± 0.8	15.5 ± 0.8	14.8 ± 1.4	15.7 ± 1.2
HR (bpm)	467.4 ± 8.5	455.6 ± 4.3	436.4 ± 9.7	454.5 ± 18.6
n (number of mice)	14	17	9	9

IVDd: intraventricular septum thickness in diastole, LVIDd: left ventricular internal diameter in diastole, LVPWd: left ventricular posterior wall thickness in diastole, LV mass/BW: ratio of left ventricular mass to body weight, FS: fractional shortening, EF: ejection fraction, HR: heart rate. Means ± SE.

**Table 2 cells-10-00535-t002:** Echocardiographic parameters measured in E1-CAAX mice 8 weeks after Sham or TAC surgery.

Parameter	Sham	TAC
Pressure gradient (mmHg)	4.3 ± 0.4	63.7 ± 7.6 *
IVDd (mm)	1.01 ± 0.08	1.25 ± 0.04 *
LVIDd (mm)	3.10 ± 0.26	3.34 ± 0.16
LVPWd (mm)	1.20 ± 0.13	1.35 ± 0.10
LV mass/BW (mg/g)	4.89± 0.40	7.12 ± 0.43 *
FS (%)	43.4 ± 6.2	32.5 ± 2.7
EF (%)	73.6 ± 5.6	61.6 ± 4.0
Stroke volume (μL)	30.2 ± 5.6	29.1 ± 3.3
Cardiac Output (mL/min)	15.8 ± 2.1	13.6 ± 1.7
HR (bpm)	463.8 ± 21.8	466.5 ± 6.9
n (number of mice)	5	6

IVDd: intraventricular septum thickness in diastole, LVIDd: left ventricular internal diameter in diastole, LVPWd: left ventricular posterior wall thickness in diastole, LV mass/BW: ratio of left ventricular mass to body weight, FS: fractional shortening, EF: ejection fraction, HR: heart rate. Means ± SE. * Statistically significant differences at *p* < 0.05 by one-way ANOVA.

## Data Availability

All data are contained within the article or Appendix A.

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
