# Peer review of "Cardiac Hypertrophy Changes Compartmentation of cAMP in Non-Raft Membrane Microdomains"

_cells, 2021, doi:10.3390/cells10030535_

Round 1

Reviewer 1 Report

Compartmentalization of cAMP within cardiomyocytes (CMs) has been suggested to regulated cellular response to injury and this group previously showed differential compartmentalization of cAMP to caveolae domains following TAC (Perera et al, Circ Res, 2015). Here, they have developed a new mouse model to detect differential cAMP compartmentalization in non-caveolar sarcolemmal microdomains. They first show generation of the E-CAAX mice and demonstrate that these mice show a healthy basal cardiac phenotype and distribution of the E1-CAAX cAMP sensor is not affected by TAC. The authors then go on to characterize cAMP accumulation in these microdomains following beta-AR and prostaglandin stimulation and the relative contribution of PDE2, 3, and 4. This is a nice piece of scientific work, though the general impact of their results to the field is a little unclear. The rationale for undertaking the study is well laid out, but this manuscript could be improved by an enhanced discussion of the impact of these results and how they contribute to our understanding of cardiac pathology.

Specific comments:

  • A missing piece of data appears to be demonstration that the E1-CAAX have a similar response to TAC as wild-type/controls (littermate controls best). The results are interesting, but proper interpretation of these results is dependent on these mice having a “normal” pathological response to TAC.
  • Results need to be described better in the text of the manuscript (e.g. results section). The figure legends are thorough for interpretation, but there appears to be much more data in the manuscript that is not discussed in detail within the text itself. Similarly, discussion of the impact of these results and how they contribute to our understanding of cardiac pathology should be improved as noted above.
  • The effect of beta1 vs beta2 ARs are considered independently, but not beta3 receptors, which have been suggested to be expressed in cardiac myocytes.

Author Response

Comments and Suggestions for Authors reviewer 1

Compartmentalization of cAMP within cardiomyocytes (CMs) has been suggested to regulated cellular response to injury and this group previously showed differential compartmentalization of cAMP to caveolae domains following TAC (Perera et al, Circ Res, 2015). Here, they have developed a new mouse model to detect differential cAMP compartmentalization in non-caveolar sarcolemmal microdomains. They first show generation of the E-CAAX mice and demonstrate that these mice show a healthy basal cardiac phenotype and distribution of the E1-CAAX cAMP sensor is not affected by TAC. The authors then go on to characterize cAMP accumulation in these microdomains following beta-AR and prostaglandin stimulation and the relative contribution of PDE2, 3, and 4. This is a nice piece of scientific work, though the general impact of their results to the field is a little unclear. The rationale for undertaking the study is well laid out, but this manuscript could be improved by an enhanced discussion of the impact of these results and how they contribute to our understanding of cardiac pathology.

Specific comments:

  • A missing piece of data appears to be demonstration that the E1-CAAX have a similar response to TAC as wild-type/controls (littermate controls best). The results are interesting, but proper interpretation of these results is dependent on these mice having a “normal” pathological response to TAC.

We thank the Reviewer for raising this point. In the first version of the manuscript, at the place where we say that E1-CAAX mice responded to TAC in a similar way as wildtypes (lines 222-225 and 405), we were citing some of our own literature which has described the wildtype FVB/N mouse phenotype in the same model 8 weeks after TAC. When performing the current study we have not TACed wildtype control littermates in parallel for animal welfare reasons and also because the compensated wildtype FVB/N mouse phenotype is very well documented in the literature. In contrast to e.g. C57Bl6/J and especially to C57Bl6/N mice, FVB/N animals do develop cardiac hypertrophy but do not show any signs of LV dilation and any major reduction in EF or FS % even 8 weeks after TAC. Overall, the response of E1-CAAX mice to TAC equals the response of wildtype FVB/N mice. We have now cited some more studies which have documented the FVB/N mouse phenotype by echocardiography performed 6-8 weeks after TAC:

https://pubmed.ncbi.nlm.nih.gov/19060905/

https://www.ncbi.nlm.nih.gov/pmc/articles/PMC3534857/

https://journals.plos.org/plosone/article?id=10.1371/journal.pone.0053951 (up to 13 weeks after TAC)

Please, see new additional references 24, 25, and 26 which were included.

Just to give you another example, here are the data from the wildtype cohort of mice which we TACed in parallel to this study (equally showing compensated hypertrophy phenotype):

  • Results need to be described better in the text of the manuscript (e.g. results section). The figure legends are thorough for interpretation, but there appears to be much more data in the manuscript that is not discussed in detail within the text itself. Similarly, discussion of the impact of these results and how they contribute to our understanding of cardiac pathology should be improved as noted above.

We thank the Reviewer for this important suggestion! We have now expanded the description of the results throughout the whole section and worked much more on expanding the discussion to include better explanation of the main results of the study and bring them into context of the current literature and understanding of cardiac pathophysiology. Please see the new text highlighted in red (track changes mode) on page 12. We also included a new Figure 8 to graphically summarize the main findings.

  • The effect of beta1 vs beta2 ARs are considered independently, but not beta3 receptors, which have been suggested to be expressed in cardiac myocytes.

Thank you for this interesting point! Indeed, beta3AR have been shown to be expressed in mouse and rat myocytes where they are preferentially coupled to Gi (in constrat to adipose tissue where this receptor stimulates cAMP production via Gs), and also to cGMP prodution which can even be measured by FRET (see e.g. Schobesberger et al 2020, https://pubmed.ncbi.nlm.nih.gov/32228862/). At the beginning of our current project we tested the effect of ISO+CGP+ICI on E1-CAAX signal in myocytes and saw zero response. Therefore, we did not follow up on beta3AR signaling in this project.

Reviewer 2 Report

This manuscript introduces a novel transgenic mouse line expressing non lipid raft membrane-targeted FRET biosensor that improves the study of cAMP microdomains at the plasma membrane within adult left ventricular myocytes. This allows live cell imaging in freshly isolated cardiomyocytes overcoming the cell infection and the ex vivo culture,  and to study cAMP alterations in vivo models of cardiac disease.  Using a well-established model of cardiac hypertrophy following TAC, the authors demonstrate that during early cardiac hypertrophy some disease-driven changes occurred in the non-raft sarcolemmal cAMP microdomains. In TAC animal those modifications led to an increase in cAMP accumulation in response to β2-AR and Prostaglandin E1-R stimulation, but not β1-AR stimulation. This was complemented by a reduction only in the PDE3 activity following β1- and β2-AR activation by submaximal and maximal isoproterenol dose respectively. No changes in PDEs regulation of Prostaglandin E1-R/cAMP signaling. Interestingly the results exhibit new insights into regulation of cAMP signaling at specific membrane microdomains during the early stage of cardiac remodelling, however they do not  provide further investigations or potential therapeutic strategies.

To strengthen the revised manuscript in terms of conclusions and data presentation, the authors should refer to the concerns listed below.

Minor points:

  1. It is not reported if the cardiomyocytes from each animal, (Sham and TAC) have been tested the same day with each drugs (ISO, PGE1, individual PDE inhibitor), to generate a set of data per day and confirm the trend. This because the FRET traces looks quite different, some of them show a “very contacting cell” (see for example Fig.4a vs 4b and 4c).
  2. Line 89 the reference is not reported as it should be and it is not present in the references list.
  3. I suggest to change cardiomyocytes (CMs) with adult mouse ventricular myocytes (AMVMs) to be  more precise.
  4. Check the first letter for the word “Sham”, as sometime is lowercase sometime is uppercase without a particular reason, for example line 91 and line 97.
  5. In Fig.5 you show the response of selective PDE inhibition after “submaximal β1-AR stimulation”, but in the legend is reported 100nmol/L ISO (line 267)

Major points:

  1. In adult ventricular myocytes, basal levels of cAMP are actually high in the bulk cytosol as well as in subcellular domains associated with the plasma membrane. Differences in adenylyl cyclase activity may contribute to some cAMP signaling modifications at no lipid raft microcompartment in TAC animals. It should be investigated further, possibly with FRET experiments, using the irreversible inhibitor of AC activity.
  2. The protein level of Prostaglandin E receptors should be tested in Sham and TAC animals.
  3. As the major point is the different PDE dependent shaping of non-raft sarcolemmal cAMP microdomains in hypertrophied CM, authors should investigate the relative contribution of each of PDE isoforms to cAMP responses associated with non-raft domains at basal condition and if this contribution changes in the first stage of cardiac remodelling.
  4. As the effect of PDE3 inhibitions as revealed only after submaximal β1-AR stimulation, it should be used also a submaximal β2-AR stimulation, to exclude no effect of PDE2 inhibitions.
  5. There is a big discrepancy between the values of cAMP response to β-ARs and prostaglandin stimulation (fig 1b) and the representative traces (fig 6a-b, fig7a-b-c). In graph 3b is shown a significant increase of cAMP level upon β2-AR stimulation in TAC, but in two of the three representative traces in fig 6 (chosen by authors) the FRET change is higher in Sham upon β2-AR stimulation. Likewise In graph 3b is shown a significant increase of cAMP level upon prostaglandin stimulation in TAC, but all three representative traces in fig 7 the FRET change is higher in Sham upon prostaglandin Are those values included in graph 3b? The traces should be representative of all data shown. Moreover few FRET traces do not show a plateau of the response after a drug is added, that is important to not misinterpreting the drug dependent effect (like fig5 b, fig6 a fig7 a-b-c).
  6. The authors can try to include a general figure/cartoon depicting the overall concept. Possible future perspectives is important to conclude the manuscript on an high note.

Author Response

Comments and Suggestions for Authors reviewer 2

This manuscript introduces a novel transgenic mouse line expressing non lipid raft membrane-targeted FRET biosensor that improves the study of cAMP microdomains at the plasma membrane within adult left ventricular myocytes. This allows live cell imaging in freshly isolated cardiomyocytes overcoming the cell infection and the ex vivo culture,  and to study cAMP alterations in vivo models of cardiac disease.  Using a well-established model of cardiac hypertrophy following TAC, the authors demonstrate that during early cardiac hypertrophy some disease-driven changes occurred in the non-raft sarcolemmal cAMP microdomains. In TAC animal those modifications led to an increase in cAMP accumulation in response to β2-AR and Prostaglandin E1-R stimulation, but not β1-AR stimulation. This was complemented by a reduction only in the PDE3 activity following β1- and β2-AR activation by submaximal and maximal isoproterenol dose respectively. No changes in PDEs regulation of Prostaglandin E1-R/cAMP signaling. Interestingly the results exhibit new insights into regulation of cAMP signaling at specific membrane microdomains during the early stage of cardiac remodelling, however they do not  provide further investigations or potential therapeutic strategies.

To strengthen the revised manuscript in terms of conclusions and data presentation, the authors should refer to the concerns listed below.

We would like to thank the reviewer for the positive evaluation of our submission and for the helpful suggestions.

Minor points:

  1. It is not reported if the cardiomyocytes from each animal, (Sham and TAC) have been tested the same day with each drugs (ISO, PGE1, individual PDE inhibitor), to generate a set of data per day and confirm the trend. This because the FRET traces looks quite different, some of them show a “very contacting cell” (see for example Fig.4a vs 4b and 4c).

This was indeed the case. We have added this information in the Methods sections, please see lines 135-136: “AMVMs from each animal have been tested on the same day with each drugs to better compare their effects.

  1. Line 89 the reference is not reported as it should be and it is not present in the references list.

We have inserted this reference into the list – now Ref 19

  1. I suggest to change cardiomyocytes (CMs) with adult mouse ventricular myocytes (AMVMs) to be  more precise.

Changed CMs to AMVMs as suggested throughout the manuscript

  1. Check the first letter for the word “Sham”, as sometime is lowercase sometime is uppercase without a particular reason, for example line 91 and line 97.

Thank you for this comment. We have changed sham to Sham throughout

  1. In Fig.5 you show the response of selective PDE inhibition after “submaximal β1-AR stimulation”, but in the legend is reported 100nmol/L ISO (line 267)

We have checked this wrong info from 100 to 3 nmol/L as appropriate

Major points:

  1. In adult ventricular myocytes, basal levels of cAMP are actually high in the bulk cytosol as well as in subcellular domains associated with the plasma membrane. Differences in adenylyl cyclase activity may contribute to some cAMP signaling modifications at no lipid raft microcompartment in TAC animals. It should be investigated further, possibly with FRET experiments, using the irreversible inhibitor of AC activity.

We fully agree with the Reviewer that ACs play an important role in the regulation of local cAMP. Ref 15 has even found stronger effects of the AC inhibitor MDL in Epac2-CAAX biosensor expressing rat myocytes when compared to cytosolic sensor or the pmEpac2 construct targeted to caveolin-rich membrane. Therefore we now included the data on the magnitude of the MDL effect measured in E1-CAAX myocytes into the Supplementary Figure 1 legend. These data could be extracted from the experiments done in Supplementary Figure 1. We found a »12% decrease of CFP/YFP ratio upon MDL treatment, somewhat similar to strong MDL effect described in Ref 15.  However, we could not repeat the same measurements in TAC myocytes since no operated mice were available in the short period of time allocated for the revision.

  1. The protein level of Prostaglandin E receptors should be tested in Sham and TAC animals.

Many thanks for this nice idea! As suggested by the Reviewer we performed immunoblot analysis of the EP4 receptor expression in Sham vs TAC heart lysate and found a clear upregulation of the receptor expression after TAC which can now help us explain the stronger PGE1 signal in the absence of any changes in PDE regulation.

Please see the New Supplementary Fig. 5 and the new text on

Lines 371-374: ”Possible explanation for this results could be an increased prostaglandin receptor ex-pression after TAC. Indeed, immunoblots analysis with a specific EP4 receptor antibody revealed its clear upregulation in TAC heart lysates (Supplementary Fig. 5)”

  1. As the major point is the different PDE dependent shaping of non-raft sarcolemmal cAMP microdomains in hypertrophied CM, authors should investigate the relative contribution of each of PDE isoforms to cAMP responses associated with non-raft domains at basal condition and if this contribution changes in the first stage of cardiac remodelling.

We thank the Reviewer for raising this important point. To assess the contribution of individual PDEs at basal, we performed additional experiments where we stimulated E1-CAAX myocytes with BAY, Cilo or Roli alone followed by IBMX and Forskolin. Surprisingly we could not detect a strong response to any selective PDE inhibitor applied alone (all »10% changes of FRET from max response). This is in contrast to previously documented strong responses measured in rat myocytes using Epac2-camps-CAAX biosensor. The reason for this discrepancy could be the use of mouse vs rat cells and Epac1-CAAX vs Epac2-CAAX sensor, because the latter has roughly two time lower EC50 values than our current sensor so can potentially respond stronger to lower amounts of cAMP. Since the detected basal responses were so small, we have not performed the same measurements after TAC because no major alterations/insights can be expected from such measurements.

These basal data can be found in the New Supplementary Fig. 4

We also added a new text to describe these considerations in the result section:

Lines 263-265: «Since cAMP signals induced by individual PDE inhibitors applied alone were negligible in E1-CAAX expressing cells (Supplementary Fig. 4), we further analyzed their effects after activation of individual cAMP stimulating receptor subtypes.»

And in the discussion, please see

Lines 433-438: “Interestingly, we did not observe strong effects of PDE inhibitors applied alone, which is in contrast to the data obtained in rat cardiomyocytes using adenoviraly expressed slightly more sensitive Epac2-CAAX biosensor [15]. However, we could detect a strong increase of local cAMP upon IBMX treatment (Supplementary Fig. 4), suggesting that multiple PDEs act in concert to control basal cAMP levels in non-caveolar membrane microdomains.”

  1. As the effect of PDE3 inhibitions as revealed only after submaximal β1-AR stimulation, it should be used also a submaximal β2-AR stimulation, to exclude no effect of PDE2 inhibitions.

Thank you for this suggestion! We have now measured the effects of PDE inhibitors on cAMP levels after submaximal beta2AR stimulation (3 nM ISO plus 100 nM CGP). Also the cAMP output was reduced, the relative contributions of individual PDEs remained roughly the same. Please, compare these new data to sham bars in the figure 6d. There was no obvious change in PDE2 inhibitor effect. Therefore, we decided not to include beta2 submax data into the manuscript.

  1. There is a big discrepancy between the values of cAMP response to β-ARs and prostaglandin stimulation (fig 1b) and the representative traces (fig 6a-b, fig7a-b-c). In graph 3b is shown a significant increase of cAMP level upon β2-AR stimulation in TAC, but in two of the three representative traces in fig 6 (chosen by authors) the FRET change is higher in Sham upon β2-AR stimulation. Likewise In graph 3b is shown a significant increase of cAMP level upon prostaglandin stimulation in TAC, but all three representative traces in fig 7 the FRET change is higher in Sham upon prostaglandin Are those values included in graph 3b? The traces should be representative of all data shown. Moreover few FRET traces do not show a plateau of the response after a drug is added, that is important to not misinterpreting the drug dependent effect (like fig5 b, fig6 a fig7 a-b-c).

We would like to thank the Reviewer for this important point. Indeed, some of the originally included traces were not representative and/or showed photobleaching. We have now improved the presentation by including new traces in Figures 7a,b,c; 6a,b; 5a,b,c as suggested.

  1. The authors can try to include a general figure/cartoon depicting the overall concept. Possible future perspectives is important to conclude the manuscript on an high note.

Thank you for this suggestion! We have now included a new Figure 8  a schematic which helps us to better summarize the main results of the study. We also extended the discussion of the paper a bit to help better explain them- please see the new text highlighted in red (track changes mode) on page 12.

Round 2

Reviewer 2 Report

The authors have satisfactorily responded to all my questions and made the necessary changes to the manuscript.